# Pulpal Response to the Combined Use of Mineral Trioxide Aggregate and Iloprost for Direct Pulp Capping

**AlAnoud Almeshari [1], Rita Khounganian [2], Wael Mahdi [3], Fahd Aljarbou [1], Shilpa Bhandi [4] and Sara Alsubait [1,***

[1] Department of Restorative Dental Sciences, College of Dentistry, King Saud University, Riyadh 11612, Saudi Arabia; alanoud.almeshari@gmail.com (A.A.); faljarbou@ksu.edu.sa (F.A.)

[2] Department of Oral Medicine and Diagnostic Sciences, College of Dentistry, King Saud University, Riyadh 11545, Saudi Arabia; ritak@ksu.edu.sa

[3] Department of Pharmaceutics, College of Pharmacy, King Saud University, Riyadh 11451, Saudi Arabia; wmahdi@ksu.edu.sa

[4] Department of Restorative Dental Sciences, College of Dentistry, Jazan University, Jazan 45142, Saudi Arabia; shilpa.bhandi@gmail.com

* Correspondence: salsubait@ksu.edu.sa

**Abstract:** Purpose: The present study aims to assess the combined effects of mineral trioxide aggregate (MTA) and iloprost when used as a pulp capping material on pulpal inflammation and tertiary dentin formation compared with MTA and iloprost alone in rat molar teeth. Methods: Eighty maxillary first molar rat teeth were exposed and capped with iloprost solution, MTA, or MTA mixed with iloprost (MTA-iloprost). The cavities were then filled with resin-modified glass ionomer. The cavity was restored with glass ionomer without the use of pulp capping agent in the control group. The rats were sacrificed after one and four weeks. Block sections of the molar specimens were prepared and subjected to hematoxylin and eosin staining for evaluation. Statistical analysis was done using the Kruskal–Wallis test, followed by Dunnett's test. Results: At week one, the control group showed significantly more severe pulpal inflammatory reactions than the iloprost ($p = 0.00$), MTA ($p = 0.04$), and MTA-iloprost ($p = 0.00$) groups. Hard tissue formation was commonly found in the iloprost, MTA, and MTA-iloprost groups. After four weeks, pulpal tissue degeneration was observed in the control group. Complete hard tissue barriers were found in 50%, 72.7%, and 77.8% of the specimens in iloprost, MTA, and MTA-iloprost groups, respectively, with no significant differences among the experimental groups. The dentinal tubule patterns were mostly regular in the MTA-iloprost group and irregular in the iloprost and MTA groups. Conclusions: The application of iloprost, MTA, and MTA-iloprost as a pulp capping material resulted in similar pulpal responses in the mechanically exposed pulp of rat molars. Therefore, mixing MTA with iloprost might not be clinically significant.

**Keywords:** iloprost; inflammation; mineralization; pulp capping material; mineral trioxide aggregate; rat dental pulp

## 1. Introduction

Dental pulp is a richly vascularized connective tissue that is surrounded by a mineralized tooth structure. Vital pulp therapy (VPT) intends to preserve and maintain the vitality and function of compromised dental pulp by facilitating reparative dentin formation [1]. The success of this therapy depends on the oxygen and nutrition supply and on the recruitment of progenitor cells to dental pulp in a process called angiogenesis [2]. Angiogenesis is the process by which new blood vessels form from pre-existing capillaries. This vascular formation is pivotal in the healing sequence of dental pulp that requires hard tissue formation [3]; thus, bioactive molecules that upregulate angiogenesis could accelerate tertiary dentin bridge formation.

Iloprost is an exogenous prostacylin that enhances angiogenesis [4]. It is commonly used in the treatment of pulmonary arterial hypertension [5] and Raynaud's disease [6].

More recently, iloprost has been introduced as a potential agent for direct pulp capping [7]. Iloprost upregulates the mRNA expression of vascular endothelial growth factor (VEGF), fibroblast growth factor-2, and platelet-derived growth factor in human dental pulp stem cells [4]. In addition, it enhances tertiary dentin formation and pulpal blood flow in a mechanical exposure model in a rat molar [4,7]. Furthermore, the angiogenic potential of iloprost was confirmed in a more recent study when a tooth slice organ culture system was used for evaluation [8]. However, the requirement of a delivery system was noted.

Mineral trioxide aggregate (MTA) is a calcium silicate-based material that is associated with high success rates when used as a pulp capping material [9–11]. Although MTA is biocompatible, has good sealing ability, and is capable of stimulating tertiary dentin formation [12], several attempts have been made to improve its biological properties by mixing it with various additives. The addition of growth hormone [13], osteostatin [14], platelet-rich fibrin [15], or human placental extract [16] has been shown to enhance the proliferation and differentiation in human dental pulp stem cells compared to MTA alone. Recently, it has been found that mixing MTA with iloprost enhances the differentiation potential of mesenchymal stem cells compared with MTA mixed with distilled water (Almeshari et al., 2021, unpublished data). Thus, we suggest that iloprost could be mixed with MTA in the clinic to improve the outcome of direct pulp capping procedures. Based on previous in vitro findings, the current study was conducted to assess the combined effects of MTA and iloprost when used as a pulp capping material on pulpal inflammation and tertiary dentin formation compared with MTA and iloprost alone in an in vivo experiment in rat molar teeth.

## 2. Materials and Methods

This study was carried out in agreement with the protocols of King Saud University, Riyadh, Saudi Arabia. All animal procedures and preoperative animal care protocols were performed according to the National Institutes of Health guidelines for the care and use of laboratory animals (NIH publication #85-23 Rev. 1985). The proposal was permitted by the Research Ethics Committee of King Saud University (KSU-SE-19-64) and the College of Dentistry Research Center of King Saud University (PR 0094) in Riyadh, Saudi Arabia.

### 2.1. Sample Size Estimation

The sample size calculation was performed using G*Power 3.1.9.4 software based on an effect size of 0.91 [17], $\alpha$ error of 0.05, and power of 80%. A minimum sample size of 32 rats was sufficient to detect significant differences between the groups. When a 25% dropout was included, the required sample size was 40 rats.

### 2.2. Sample Selection, Animal Preparation, Randomization, and Grouping

A total of 80 maxillary first molars from 40 male Wistar rats weighing between 220–260 g at seven weeks of age were used. The teeth were randomly distributed in four groups according to the pulp capping material used: MTA, iloprost, MTA-iloprost, and untreated control. The maxillary first molar of each side was used for a different group. Pulpal responses were observed at two time points. The animals were placed in individual ventilated cages with an illumination cycle of light/dark every 12 h, a temperature set at $20 \pm 2\ °C$, and 60% humidity. The animals were housed under the supervision of veterinary specialists and had ad libitum access to food and water throughout the study period.

### 2.3. Surgical Procedures

Anesthetization of rats was performed with an intraperitoneal injection of 50% ketamine (Tekam, London, UK) at a dose of 100 mg/kg and 2% xylazine (Seton, Barcelona, Spain) at a dose of 10 mg/kg. Afterward, disinfection of the maxillary first molars was done with cotton soaked in 75% ethanol. Under a 4X microscope (System Contraves & Sec, Carl Zeiss, Thornwood, NY, USA), a class I cavity was prepared occlusally to reach the pulp chamber without tissue removal using a high-speed sterile no. 1/2 round carbide bur

(Komet Dental, Lemgo, Germany) with sterile saline as the coolant. The exposed area was irrigated with sterile normal saline solution. Bleeding was controlled by pressing cotton pellets soaked in room-temperature sterile saline for 10–15 s. In the experimental groups, MTA, iloprost, or MTA-iloprost was placed on the exposed pulps. MTA powder (ProRoot; Dentsply, Tulsa, OK, USA) in the MTA group was mixed with distilled water according to the manufacturer's instructions. For the MTA-iloprost group, the MTA powder was mixed with iloprost solution (MedChem Express, NJ, USA) at a concentration of $10^{-6}$ mol/L [7]. In the iloprost group, 5 μL of iloprost was injected directly into the area of pulp exposure.

Subsequently, the cavity was directly filled with resin-modified glass ionomer (GC Fuji II LC; GC America Inc, Alsip, IL, USA). The cavity was restored with glass ionomer without the use of pulp capping agent in the untreated control group. Reduction of cusp tips of the opposing teeth were done to lessen occlusal forces. The surgical procedures were all performed by one operator.

### 2.4. Sample Preparation

After pulp treatment, the animals were sacrificed at one and four weeks [18,19] by using an overdose of CO2 gas. Block sections of the molar specimens were separated from the maxillae of all rats and subjected to hematoxylin and eosin (H&E) staining.

### 2.5. Histological Examination

The blocks were fixed with 4% paraformaldehyde for two days and demineralized using 10% EDTA/phosphate-buffered saline solution for six weeks at 4 °C. Dehydration of all specimens in graded ethanol and embedding in paraffin were performed. Five-micrometer thick mesiodistal serial sections were prepared. Slices at the level of exposure were taken, numbered, and subjected to H&E staining according to conventional pathological protocols (Baso Diagnostic Inc., Zhuhai, Guangdong, China). The sections were observed by light microscope (binocular microscope; Leitz, Laborlux S, Wetzlar, Germany) for the assessment of pulp inflammation and tertiary dentin formation. The evaluation of histologic features was done according to the criteria presented in Table 1 [18,20]. Each histologic feature was evaluated using a scoring system, with grade 1 being the best result.

**Table 1.** Histological evaluation scoring protocol of pulp response to direct pulp capping.

| Grade | Inflammatory Cell Response | Hard Tissue Formation | Quality of Dentin Formation in the Bridge |
|---|---|---|---|
| 1 | Absent or few inflammatory cells | Heavy: hard tissue deposition as complete and continuous dentin bridge | Regular pattern of tubules |
| 2 | Mild: inflammatory cells only next to dentin bridge or area of pulp exposition | Moderate: hard tissue formation as incomplete and discontinuous dentin bridge | Irregular pattern of tubules |
| 3 | Moderate: inflammatory cells are observed in the part of coronal pulp | Slight: a layer of scattered and foggy hard tissue deposition | No tubules present |
| 4 | Severe: all coronal pulp | No hard tissue deposition | |

The histological features were evaluated by two trained observers. Proper assessments were started when the two observers scored equal to or greater than 0.81 (very good) agreement. histologic sections were evaluated in a blinded manner by the examiners. A consensus was reached when a discrepancy existed between the two observers.

### 2.6. Statistical Analysis

The histologic examination results were compared using the Kruskal–Wallis test followed by Dunnett's test for analysis between groups. The Cohen kappa coefficient of the agreement index was used to assess the agreement between the two evaluators. SPSS software (version 23.0, SPSS IBM, Armonk, NY, USA) was used for statistical analysis. Statistical significance was set at 0.05.

## 3. Results

Among the 80 molars, 11 teeth were not included in the analysis, comprising four from the control group, one from the iloprost group, three from the MTA group, and three from the MTA-iloprost group. The reasons for exclusion were either because of the loss of the restoration or an error made during processing. According to the Cohen kappa statistics, the interobserver agreement was determined to be 86%. The scoring results of the histologic findings are summarized in Table 2.

**Table 2.** Histological section scoring of pulps capped with different test materials.

| Time | Group | No. of Specimen | Inflammatory Cell Response Score (%) | | | | Hard Tissue Formation Score (%) | | | | Quality of Dentin Formation Score (%) | | |
|---|---|---|---|---|---|---|---|---|---|---|---|---|---|
| | | | 1 | 2 | 3 | 4 | 1 | 2 | 3 | 4 | 1 | 2 | 3 |
| Week 1 | Control | 9 | 0.0 | 0.0 | 33.3 | 66.7 | 0.0 | 0.0 | 55.6 | 44.4 | - | - | - |
| | Iloprost | 9 | 44.4 | 55.6 | 0.0 | 0.0 | 0.0 | 44.4 | 44.4 | 11.1 | - | - | - |
| | MTA | 7 | 28.6 | 28.6 | 42.9 | 0.0 | 0.0 | 85.7 | 14.3 | 0.0 | - | - | - |
| | MTA- ILO | 8 | 37.5 | 50.0 | 12.5 | 0.0 | 12.5 | 62.5 | 25.0 | 0.0 | - | - | - |
| Week 4 | Control | 7 | 57.1 | 28.6 | 14.3 | 0.0 | 0.0 | 28.6 | 42.8 | 28.6 | 0.0 | 28.5 | 71.5 |
| | Iloprost | 10 | 50.0 | 40.0 | 10.0 | 0.0 | 50.0 | 30.0 | 20.0 | 0.0 | 10.0 | 70.0 | 20.0 |
| | MTA | 11 | 63.6 | 36.4 | 0.0 | 0.0 | 72.7 | 9.1 | 18.2 | 0.0 | 27.3 | 54.5 | 18.2 |
| | MTA- ILO | 9 | 77.8 | 22.2 | 0.0 | 0.0 | 77.8 | 0.0 | 22.2 | 0.0 | 55.6 | 22.2 | 22.2 |

### 3.1. First Week of Observation

At week one, a significant difference was observed in the inflammatory response scores between the groups ($p = 0.00$). The control group had more severe pulpal inflammatory reactions than the iloprost ($p = 0.00$), MTA ($p = 0.04$), and MTA-iloprost ($p = 0.00$) groups. Moderate to severe inflammation with areas filled with granulation tissue and degeneration was seen in the control group (Figure 1A,B). However, none of the specimens in the other groups showed a severe inflammatory response (Figure 2A–F). Furthermore, the odontoblastic layer was lost in the control group, but was mostly present in the other groups. A common finding that was observed in the iloprost- and MTA-iloprost-treated specimens was the presence of engorged and dilated blood vessels that were lined by endothelial cells (Figure 2B,F).

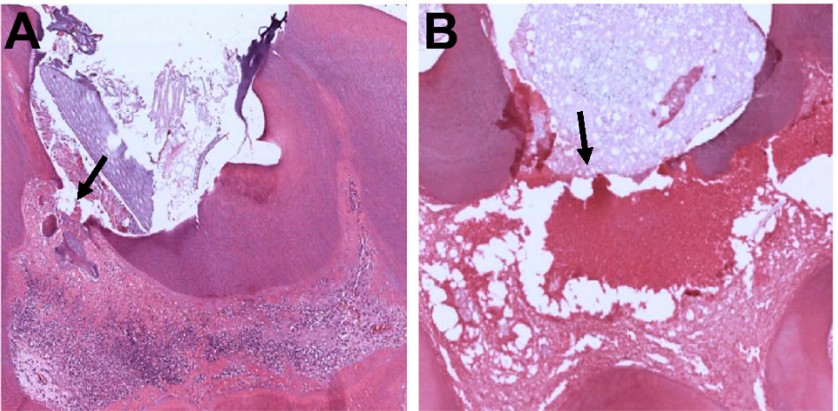

**Figure 1.** Histologic features of maxillary first molars subjected to pulp exposure and restored with glass ionomer without the use of pulp capping agent at week one. Representative photomicrographs showing (**A**) diffuse inflammation and (**B**) degenerated pulp with granulation tissue filling the exposure area. H&E stain ×200 magnification. Arrows indicate the exposure area.

Hard tissue formation was repeatedly found in the iloprost, MTA, and MTA-iloprost groups. A significant difference was detected among the groups for hard tissue formation ($p = 0.00$). The use of MTA and MTA-iloprost was associated with a higher rate of barrier formation than the control group ($p = 0.00$ and $p = 0.00$, respectively). For the iloprost group, the reparative dentin was mostly found on the lateral walls of the pulp chamber, rather than at the exposure site (Figure 2A,B).

### 3.2. Four-Week Observation

After four weeks, the majority of specimens showed a virtual absence or the presence of only a few inflammatory cells. No significant difference was found among the groups in terms of inflammatory response ($p = 0.24$). Pulpal tissue degeneration was observed in the control group (Figure 3A). Incomplete dentin bridges or distributed hard tissue were observed in 42.8% of the control group specimens (Figure 3B). On the other hand, a complete hard tissue barrier was found in 50%, 72.7%, and 77.8% of the iloprost, MTA, and MTA-iloprost groups, respectively (Figure 4A–C). A significant difference was detected among the groups in terms of hard tissue formation ($p = 0.01$). The use of MTA and MTA-iloprost was coupled with a higher rate of barrier formation than that observed in the control group (MTA vs. control $p = 0.01$ and MTA-iloprost vs. control $p = 0.01$, respectively). Moreover, the groups showed a significant difference in the quality of the newly formed calcified bridge ($p = 0.04$).

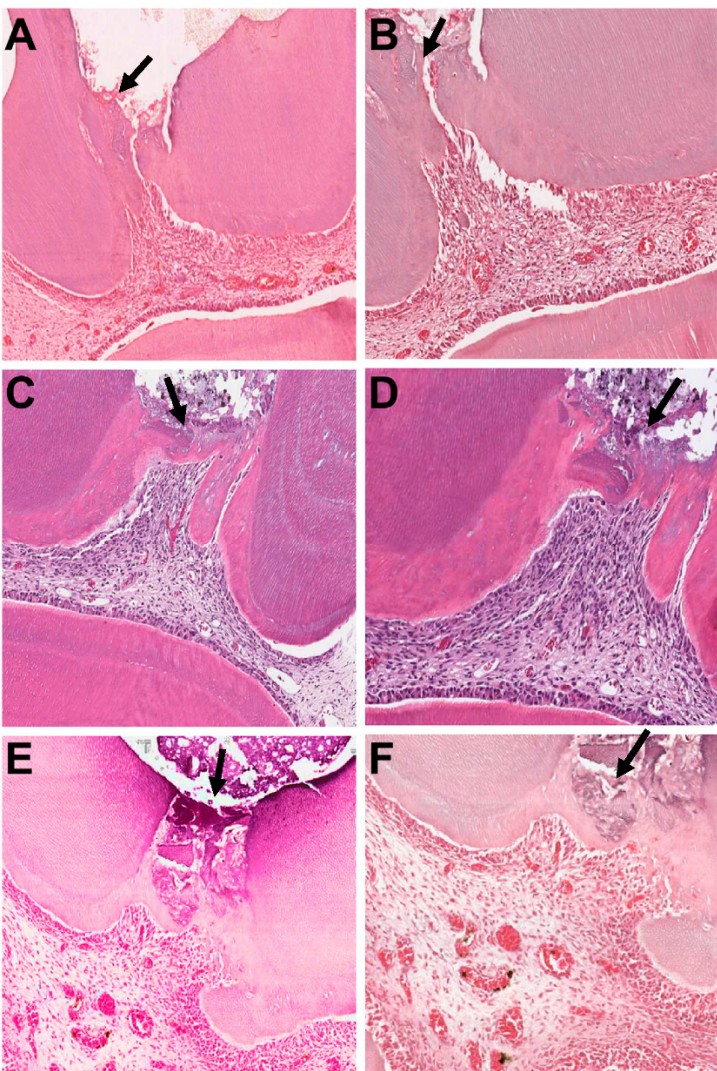

**Figure 2.** Histologic features of maxillary first molars capped with (**A**,**B**) iloprost, (**C**,**D**) MTA, and (**E**,**F**) MTA-iloprost at one week. Higher magnification showing dilated and engorged blood vessels in (**B**) iloprost and (**F**) MTA-iloprost specimens. H&E stain ×200 and ×400 magnification. Arrows indicate tertiary dentin.

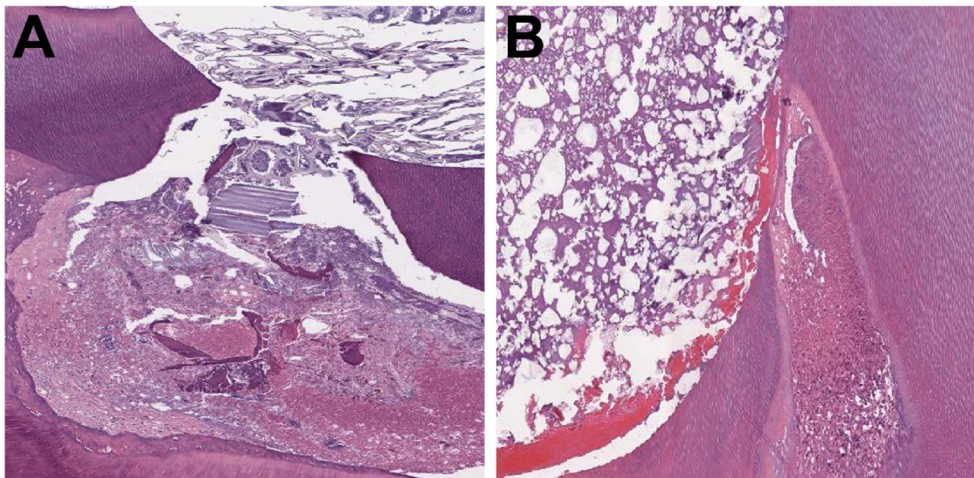

**Figure 3.** Histologic features of maxillary first molars subjected to pulp exposure and restored with glass ionomer without the use of pulp capping agent at week four. Representative photomicrographs showing (**A**) pulpal degeneration and (**B**) calcified bridge formation. H&E ×200 magnification.

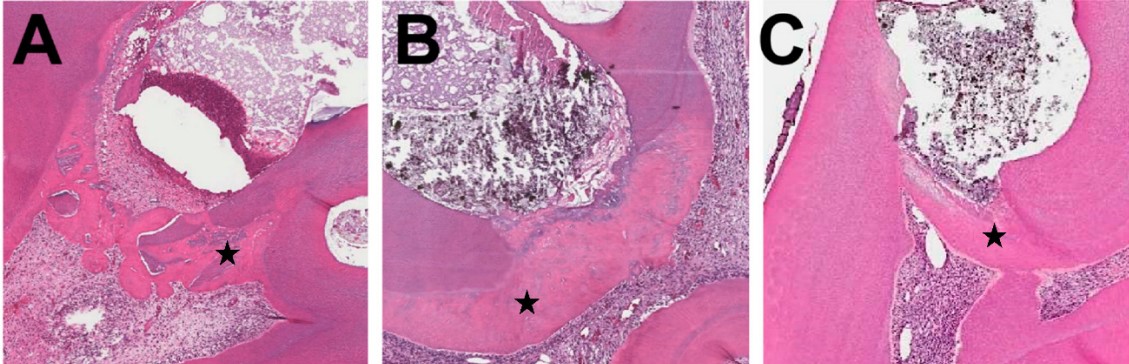

**Figure 4.** Histologic features of maxillary first molars subjected to pulp exposure and capped with (**A**) iloprost, (**B**) MTA, and (**C**) MTA-iloprost at week four. H&E stain ×200 magnification. The star indicates tertiary dentin.

The hard tissue barrier in the majority of specimens in the MTA-iloprost group showed a regular pattern of tubules (Figure 4C). On the other hand, the majority of specimens in the iloprost and MTA groups had irregular patterns of dentinal tubules (Figure 4A,B). A statistically significant —difference was observed in the quality of newly formed hard tissue between the control group and MTA ($p$ = 0.02), and between the control group and MTA-iloprost ($p$ = 0.01).

## 4. Discussion

The preclinical assessment of a modified dental material has to be experimentally evaluated to determine its biological behavior prior to its clinical application. The use of in vivo models provides a complex systemic interaction between the biomaterial and the underlying different cell types. Rat molars have been shown to be valid experimental models for evaluating pulpal reactions after capping with different biomaterials [7,19,21]. Rats have a similar oral bacterial flora to that of humans [22]. Rat molars are anatomically and histologically similar to human molar teeth [23]. Furthermore, the healing process after direct pulp capping in rats is comparable to that observed in humans [24]. Dentinal bridge formation in rats can be shown histologically within four weeks after direct pulp capping [18,19], making the rat a suitable model for short-term evaluations. Two different time points (one and four weeks) were selected in the present study to assess the effect of

the combined use of MTA and iloprost on the progress of the pulpal healing process, in accordance with previous studies [18,19].

MTA is a bioactive material that is biocompatible, radiopaque, has low solubility, and induces dentin bridge formation. Like any other material, MTA has a number of drawbacks. It has a long setting time, high cost, and potential of discoloration. The metal oxide content of MTA has been reported as the cause of discoloration [25]. Several methods have been introduced to overcome it, such as sealing dentinal tubules of the pulp chamber by bonding agent before MTA application [26], placing MTA below the cementoenamel junction, and using MTA when esthetic is not a concern, like in posterior teeth. MTA is considered the gold standard among other calcium silicate-based materials, and that is why it was selected in our study. Moreover, MTA's physical and chemical properties have been shown to be affected by the acidic environment [27,28], therefore, a composite was avoided and resin-modified glass ionomer was used to cover MTA.

In the present study, a nontreated control group was included to differentiate between the self-repair and material-induced repair processes. At week one, all teeth in the control group showed moderate to severe pulpal inflammation, which was not due to cavity preparation per se, whereas the other groups did not show the same degree of inflammation. However, the inflammatory response within the control group may be attributed to the cytotoxicity of the resin- modified glass ionomer [29]. At week four, slight to moderate inflammation was present with signs of pulpal degeneration, which may imply a late stage of chronic inflammation. Some of the control samples showed hard tissue formation. The formation of mineralized tissues in the control group has been previously reported in studies using rat molars [18,20]. This may be explained by the sealing ability of the resin-modified glass ionomer material [30]. Pulp has the inherent capability of healing and forming dentin bridges in the absence of bacterial microleakage [31].

Initial pulpal inflammation is a prerequisite for healing and regeneration [32]. However, promoting the resolution of the inflammatory response would favor the dental regenerative process [33]. With time, specimens in the experimental groups showed less inflammation and greater formation of continuous mineralized bridges, indicating the biocompatibility of MTA and iloprost. This finding is in agreement with previous studies that showed the formation of mineralized tissue after the use of MTA [18,20] or iloprost [7,34] to cap mechanically exposed dental pulp. Interestingly, in the iloprost group, the reparative dentin was mostly found on the lateral walls of the exposure site at week one. This could be explained by the liquid nature of iloprost, making it almost impossible to control its deposition site. Thus, MTA was mixed with iloprost to control its delivery.

This study revealed that pulp capping with either MTA or MTA-iloprost showed no significant difference in inflammatory pulpal responses and tertiary dentin formation. This could be due to the similarities in their composition, where the only difference is the presence of iloprost in the MTA-iloprost group. However, the formation of a continuous bridge was observed earlier in the MTA-iloprost group. Mineralization and angiogenesis are closely related. Iloprost is able to upregulate angiogenesis and increase VEGF expression and pulpal blood flow [4], which enhances the recruitment and differentiation of osteo/odontoprogenitor cells [4]. Additionally, VEGF was reported to induce the mineralization of bone marrow-derived mesenchymal stem cells and human dental pulp stem cells [35]. The results of the current study demonstrated that MTA mixed with iloprost had a higher angiogenic potential than MTA, as evidenced by dilated and engorged blood vessels in the histologic sections. Further studies are required to evaluate the angiogenic properties of MTA when mixed with iloprost.

Another factor that was evaluated in this study was the quality of the newly formed calcified bridge. The majority of specimens in the MTA-iloprost group had a regular pattern of tubules, while the MTA group showed an irregular pattern of tubules. A possible explanation for this observation is that tertiary dentin formation often starts as irregular and/or atubular dentin, and then the formation of a tubular dentin-like matrix takes place later by the more palisaded odontoblast-like cells [36]. This confirms that the formation

of dentin bridges in the MTA-iloprost group starts earlier than in the MTA group, and it achieves more regular tubularization. In our previous in vitro experiment, it was found that MTA mixed with iloprost upregulated the expression of differentiation markers, including bone sialoprotein (BSP) and osteopontin (OSP), compared with MTA (Almeshari et al., 2021, unpublished data). The ability of BSP to induce the formation of homogenous reparative mineralized tissue when implanted directly on mechanically exposed pulp has been reported in a previous study [37], thus suggesting its role during reparative dentinogenesis. OSP is another hard tissue-related protein that is believed to have some role in the initiation of the reparative process by its deposition in the superficial layer in the exposed pulpal matrix, followed by the attraction and differentiation of odontoblast-like cells [38]. Further research is required to study the mechanism of the signaling processes when MTA is mixed—with iloprost. Nonetheless, pulp capping therapy is performed when pulp exposure occurs due to caries, trauma, or iatrogenic insult. The main limitation of the present study was that capping was performed immediately after exposure of normal pulp tissue; thus, it does not represent all clinical scenarios. Further in vivo studies on inflamed pulps are necessary to investigate the potential application of the combination of MTA and iloprost as pulp capping material.

**5. Conclusions**

The present in vivo study results demonstrate that iloprost, MTA, and MTA-iloprost all have desirable outcomes when used as pulp capping materials. Biocompatibility and hard tissue formation were observed in all materials. Therefore, mixing MTA with iloprost might not be clinically significant.

**Author Contributions:** Conceptualization, A.A. and S.A.; methodology, A.A., W.M., F.A.; validation, S.B. and S.A.; formal analysis, A.A. and S.A.; investigation, A.A. and R.K.; writing—original draft preparation, A.A. and S.A.; writing—review and editing, R.K., W.M., F.A., and S.B.; visualization, S.B.; supervision, S.A.; project administration, S.A. All authors have read and agreed to the published version of the manuscript.

**Funding:** This research was supported by a Research Initiative from the Prince Naif bin Abdulaziz Health Research Center, King Saud University Medical City, Riyadh, Saudi Arabia.

**Institutional Review Board Statement:** The proposal was permitted by the Research Ethics Committee of King Saud University (KSU-SE-19-64) and the College of Dentistry Research Center of King Saud University (PR 0094) in Riyadh, Saudi Arabia.

**Informed Consent Statement:** Not applicable.

**Data Availability Statement:** The data presented in this study are available on request from the corresponding author.

**Conflicts of Interest:** The authors declare no conflict of interest.

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
