# Peer review of "Pulpal Response to the Combined Use of Mineral Trioxide Aggregate and Iloprost for Direct Pulp Capping"

_applsci, doi:10.3390/app11083702_

Round 1

Reviewer 1 Report

Dear Authors, first of all I would like to congratulate You on your work. The topic Is of great clinical relevance. However, I believe that the article could be improved. Please, take a note of some suggestions.

The introduction provides a good, generalized background of the topic that quickly gives the reader an appreciation of the wide range of problems about a dental caries. However, to make the introduction more substantial, the author may wish to provide several references to substantiate the claim made in the first sentence (that is, provide references to other groups who do or have done research in this area).

moreover, to make the motivation clearer and to differentiate the review some more from other applied papers, the author may wish to provide another sentence giving examples of some of the ideas along with appropriate references.

Section Conclusion is missing, why?

Taking everything in consideration, I strongly suggest that you rearrange the manuscript (especially Introduction and Discussion) and complete section conclusions. Please, be more focused on the topic and do not repeat the same information several times.

I believe that your manuscript would have much more relevance after suggested improvements.

Author Response

Point by Point Response

We would like to thank Editor and Reviewers for their valuable comments that helped making our manuscript better. We have added requested information and responded to all the comments. Below you will see a point by point response.

Reviewer 1:

  • Dear Authors, first of all I would like to congratulate You on your work. The topic Is of great clinical relevance. However, I believe that the article could be improved. Please, take a note of some suggestions.

The introduction provides a good, generalized background of the topic that quickly gives the reader an appreciation of the wide range of problems about a dental caries. However, to make the introduction more substantial, the author may wish to provide several references to substantiate the claim made in the first sentence (that is, provide references to other groups who do or have done research in this area).

Response:

Thank you for your comment. The following has been added:” The addition of of growth hormone [12], osteostatin [13], platelet-rich fibrin [14] or human placental extract [15] has been shown to enhance the proliferation and differentiation in human dental pulp stem cells compared to MTA alone”. Page 2. Line 61-63.

  • moreover, to make the motivation clearer and to differentiate the review some more from other applied papers, the author may wish to provide another sentence giving examples of some of the ideas along with appropriate references.

Response:

Thank you for your comment. The following has been added:” The addition of growth hormone [12], osteostatin [13], platelet-rich fibrin [14] or human placental extract [15] has been shown to enhance the proliferation and differentiation in human dental pulp stem cells compared to MTA alone”. Page 2. Line 61-63.

  • Section Conclusion is missing, why?

Thank you for your comment. The following has been added: Page 9. Line 302-307.

Response:

Conclusion

The present in vivo study results demonstrate that iloprost, MTA and MTA-iloprost all have desirable outcomes when used as pulp capping materials. Biocompatibility and hard tissue formation were observed in all materials. Therefore, mixing MTA with iloprost might not be clinically significant.

  • Taking everything in consideration, I strongly suggest that you rearrange the manuscript (especially Introduction and Discussion) and complete section conclusions.Please, be more focused on the topic and do not repeat the same information several times.

Response:

Thank you for your comment. I rearranged, deleted some points in the manuscript.

  • I believe that your manuscript would have much more relevance after suggested improvements.

Reviewer 2:

This research is under the scope of this journal; the topic is relevant for readers, and this research deals with potentially significant knowledge to the field.

However, there are many questions to be addressed, as follows:

  • Change the Type of article for Article.

Response:

Thank you for your comment. I changed the title to Article. Page 1. Line 1.

  • Abstract

In the results, is important to show more information, add some of the p-values.

Response:

Thank you for your comment. The following has been added:” The dentinal tubules pattern were mostly regular in the MTA-iloprost group and irregular in the iloprost and MTA groups.” And The p-values were added too. Page 1. Line 23-24 and 28-29.

  • The concept of work

One of the big problem of this work, how you standardized the expose! With the size of carbide bur, it saw a partial healthy pulp removal (DPC or pulpotomy procedure). Please read https://doi.org/10.1038/s41598-020-78022-w “Impact of remnant healthy pulp and apical tissue on outcomes after the simulated regenerative endodontic procedure in rat molars".

Response:

Thank you for your comment. In our experiment, no pressure was applied while using the carbide bur to approximate the pulp chamber. Single operator performed the procedure and she stopped once there was feeling of the drop and a bleeding point was shown (under microscope). There was no intention to remove healthy pulp tissue. This technique has been used in previous studies:

  1. Limjeerajarus, C.N.; Chanarattanubol, T.; Trongkij, P.; Rujiwanichkul, M.; Pavasant, P. Iloprost induces tertiary dentin formation. J Endod. 2014, 40:1784-40:1790, doi: 1016/j.joen.2014.07.002.Seang, S.; Pavasant, P.; Limjeerajarus, C.N. Iloprost induces dental pulp angiogenesis in a growth factor-free 3-dimensional organ culture system. J Endod. 2018, 44: 759-44:764, doi: 10.1016/j.joen.2018.02.001.
  2. Kim, D.H.; Jang, J.H.; Lee, B.N.; Chang, H.S.; Hwang, I.N.; Oh, W.M.; Kim, S.H.; Min, K.S.; Koh, J.T.; Hwang, Y.C. Anti-inflammatory and mineralization effects of ProRoot MTA and endocem MTA in studies of human and rat dental pulps in vitro and in vivo. J Endod. 2018, 44:1534-44:1541, doi: 10.1016/j.joen.2018.07.012.
  • (Statement of clinical Relevance): In the introduction, what is the importance of this study for the clinical?

Response:

Thank you for your comment. The following has been added (Thus, we suggest that iloprost could be mixed with MTA in the clinic to improve the outcome of direct pulp capping procedures). Page 2. Line 66 -67.

  • (M&M): To make this procedure was made under the microscope and what was the magnification used to perform the procedure?

Response:

Thank you for your comment. “The following has been added:” Under a 4X microscope”. Page 3. Line 108.

  • The authors used a carbide bur for made Direct Pulp Camping, how was it done with a turbine or by hand?  

Response:

Thank you for your comment. The carbide bur was derived by a turbine and the speed as mentioned in the methodology. Page 3. Line 110.

  • How did you expose pulp tissue? By direct contact or did you remove tissue?

Response:

Thank you for your comment. The exposure was done by direct contact. The following was added:” class I cavity was prepared occlusally to reach the pulp chamber without tissue removal”. Page 3. Line 109-110.

  • Did you use the contracting technique on all teeth subjected to pulpotomy? Or it was just to measure pulp removal.  

Response:

Can you please explain what do you mean by contracting technique? We used the bur to expose the pulp. There was no removal of pulp tissue. Teeth in the control group were included to compare the tissue reaction when left untreated with a pulp capping agent. 

  • What was the temperature of the saline solution used in cotton pellets ?

Response:

Thank you for your comment. The following was added:” pressing cotton pellets soaked in room tempreture sterile saline for 10-15 seconds”. Page 3. Line 112-113.

  • How long (minutes) did you wait before performing the restoration?

Response:

Thank you for your comment. The restoration was placed directly. The following has been added:” Subsequently, the cavity was directly filled with resin-modified glass ionomer”. Page 3. Line 120.

  • Results: On histological images, Identified the histological structures using arrows, letter…

Response:

Thank you for your comment. Arrows and stars were added as suggested.

  • Improved the quality of resolution of the histological image.

Response:

 Thank you for your comment. The resolution was adjusted.

  • (Discussion)
  • Please, clarified more limitations of this study?

Thank you for your comment. The following has been added: (Nonetheless, pulp capping therapy is performed when pulp exposure occurs due to caries, trauma or iatrogenic insult. The main limitation of the present study was that capping was performed immediately after exposure of normal pulp tissue; thus, it does not represent all clinical scenarios. Further in vivo studies on inflamed pulps are necessary to investigate the potential application of the combination of MTA and iloprost as pulp capping material). Page 9. Line 295-300.

  • I agree with the authors about the study limitation was using healthy pulp and without infection/inflammation… In the clinic, this situation was more similar to a pulpotomy technique, or a DPC, than a Vital Pulp Therapy with irreversible pulpitis.

  • Since this type of calcium silicate cement is associated with color change in the medium / long term (https://doi.org/10.3390/app10175793. Other drawbacks of MTA were a lowers values shear bond, https://doi.org/10.3390/ma11112216. How did you put the MTA? One of the major problems with MTA hydraulic cement, in addition to its setting time, is the weak connection to restorative materials, with very low values and unpredictable connections to restoration material (Palma, Materials MDPI, 2018 and Clinical Oral Invest 2020).

Response:

Thank you for your comment. Although MTA has been reported to have some limitation it is considered the gold standard material for vital pulp therapy procedures. Therefore, it was selected as a material for VPT. The information that we got from the present study can help us in developing the ideal materials for such procedures. 

  • And also, clarified the future perspectives also add in the discussion.

Response:

Thank you for your comment. The following was added:”Further in vivo studies on inflamed pulps are necessary to investigate the potential application of the combination of MTA and iloprost as pulp capping material”. Page 9. Line 298-300.

Reviewer 2 Report

This research is under the scope of this journal; the topic is relevant for readers, and this research deals with potentially significant knowledge to the field.

However, there are many questions to be addressed, as follows:

  • Change the Type of article for Article.

Abstract

  • In the results, is important to show more information, add some of the p-values.
  •  

The concept of work

One of the big problem of this work, how you standardized the expose! With the size of carbide bur, it saw a partial healthy pulp removal (DPC or pulpotomy procedure). Please read https://doi.org/10.1038/s41598-020-78022-w “Impact of remnant healthy pulp and apical tissue on outcomes after the simulated regenerative endodontic procedure in rat molars".

(Statement of clinical Relevance)

  • In the introduction, what is the importance of this study for the clinical?

(M&M)

  • To make this procedure was made under the microscope and what was the magnification used to perform the procedure?

  • The authors used a carbide bur for made Direct Pulp Camping, how was it done with a turbine or by hand?
  • How did you expose pulp tissue? By direct contact or did you remove tissue?
  • Did you use the contracting technique on all teeth subjected to pulpotomy? Or it was just to measure pulp removal.
  • What was the temperature of the saline solution used in cotton pellets
  • How long (minutes) did you wait before performing the restoration?

Results

  • On histological images, Identified the histological structures using arrows, letter…
  • Improved the quality of resolution of the histological image.

(Discussion)

  • Please, clarified more limitations of this study?
  • I agree with the authors about the study limitation was using healthy pulp and without infection/inflammation… In the clinic, this situation was more similar to a pulpotomy technique, or a DPC, than a Vital Pulp Therapy with irreversible pulpitis.
  • Since this type of calcium silicate cement is associated with color change in the medium / long term (https://doi.org/10.3390/app10175793. Other drawbacks of MTA were a lowers values shear bond, https://doi.org/10.3390/ma11112216. How did you put the MTA? One of the major problems with MTA hydraulic cement, in addition to its setting time, is the weak connection to restorative materials, with very low values and unpredictable connections to restoration material (Palma, Materials MDPI, 2018 and Clinical Oral Invest 2020).
  • And also, clarified the future perspectives also add in the discussion.

Author Response

(The authors gave the same response as above.)

Round 2

Reviewer 1 Report

Many congratulations to authors, the authors followed the suggestions of the reviewers. 

If you think it appropriate, I would like to suggest revisions regarding MTA which I report below: 

1)  Mineral trioxide aggregate applications in endodontics: A review  DOI: 10.1055/s-0040-1713073

2) Flowable resin and marginal gap on tooth third medial cavity involving enamel and radicular cementum: A SEM evaluation of two restoration techniques  DOI: 10.4103/0970-9290.111256

Author Response

Point by Point Response

We would like to thank Editor and Reviewers for their valuable comments that helped making our manuscript better. We have added the requested information and responded to all comments. Below you will see a point by point response.

Reviewer 1:

Many congratulations to authors, the authors followed the suggestions of the reviewers. 

If you think it appropriate, I would like to suggest revisions regarding MTA which I report below: 

1)  Mineral trioxide aggregate applications in endodontics: A review  DOI: 10.1055/s-0040-1713073

2) Flowable resin and marginal gap on tooth third medial cavity involving enamel and radicular cementum: A SEM evaluation of two restoration techniques  DOI: 10.4103/0970-9290.111256

Response

Thank you for your suggestion. The papers have been reviewed. Mineral trioxide aggregate applications in endodontics: A review  DOI: 10.1055/s-0040-1713073 has been added as a reference. Page 2. Line 57-58.

Reviewer 2:

The authors responded with poorly responses, see in the case of material limitations, in which the reviewer suggested that discoloration and weak adhesive bonding of CSC materials were addressed, was not taken into account. As it is something that I think is important to increase the quality of the article, I ask you to reformulate it.

Response

Thank you for your comment. The statements that we made were more ambiguous than intended, and we have added a paragraph discussing the discoloration and bond strength to make it clearer. Page 8. Line 242-252.

MTA is a bioactive material that is biocompatible, radiopaque, has low solubility and induce dentin bridge formation. Like any other material, MTA has a number of drawbacks. It has a long setting time, high cost and potential of discoloration. The metal oxide content of MTA has been reported as the cause of discoloration [25]. Several methods have been introduced to overcome it; such as sealing dentinal tubules of pulp chamber by bonding agent before MTA application(26), placing MTA below the cementoenamel junction, and using MTA when esthetic is not a concern like in posterior teeth. MTA is considered the gold standard among other calcium silicate based materials, that’s why it was selected in our study. Moreover, MTA physical and chemical properties have shown to be affected by the acidic environment(27)(28), thereby composite was avoided and resin modified glass ionomer was used to cover MTA.”

Reviewer 2 Report

The authors responded with poorly responses, see in the case of material limitations, in which the reviewer suggested that discoloration and weak adhesive bonding of CSC materials were addressed, was not taken into account. As it is something that I think is important to increase the quality of the article, I ask you to reformulate it.

Author Response

(The authors gave the same response as above.)

Round 3

Reviewer 2 Report

This review is under the scope of this journal; the topic is interesting for readers and this research deals with potentially significant knowledge to the field.

The authors improved the quality of the manuscript.